# Spontaneous and evoked neurotransmission are partially segregated at inhibitory synapses

Patricia M Horvath[1,2], Michelle K Piazza[3,4], Lisa M Monteggia[1,3]*, Ege T Kavalali[1,3]*

[1]Department of Pharmacology, Vanderbilt University, Nashville, United States; [2]Department of Neuroscience, the University of Texas Southwestern Medical Center, Dallas, United States; [3]Vanderbilt Brain Institute, Vanderbilt University, Nashville, United States; [4]Neuroscience Program, Vanderbilt University, Nashville, United States

**Abstract** Synaptic transmission is initiated via spontaneous or action-potential evoked fusion of synaptic vesicles. At excitatory synapses, glutamatergic receptors activated by spontaneous and evoked neurotransmission are segregated. Although inhibitory synapses also transmit signals spontaneously or in response to action potentials, they differ from excitatory synapses in both structure and function. Therefore, we hypothesized that inhibitory synapses may have different organizing principles. We report picrotoxin, a GABA$_A$R antagonist, blocks neurotransmission in a use-dependent manner at rat hippocampal synapses and therefore can be used to interrogate synaptic properties. Using this tool, we uncovered partial segregation of inhibitory spontaneous and evoked neurotransmission. We found up to 40% of the evoked response is mediated through GABA$_A$Rs which are only activated by evoked neurotransmission. These data indicate GABAergic spontaneous and evoked neurotransmission processes are partially non-overlapping, suggesting they may serve divergent roles in neuronal signaling.

*For correspondence:
lisa.monteggia@vanderbilt.edu
(LMM);
ege.kavalali@vanderbilt.edu (ETK)

## Introduction

Synaptic neuronal communication can be broadly classified into either evoked or spontaneous neurotransmission. Evoked neurotransmission is the canonical action-potential driven signaling that causes synchronous or asynchronous release of vesicles at multiple synapses (*Südhof, 2013*). Spontaneous neurotransmission occurs via action-potential independent release of single synaptic vesicles. At the molecular level, spontaneous neurotransmission has been shown to utilize partly different molecular machinery and act at distinct postsynaptic sites than evoked neurotransmission (*Kavalali, 2015*).

The organizing principles surrounding evoked and spontaneous neurotransmission may differ between excitatory and inhibitory synapses. Spontaneous and evoked glutamate release at excitatory synapses in the hippocampus, as well as synapses at the *Drosophila* neuromuscular junction, activate distinct sets of α-amino-3-hydroxy-5-methyl-4-isoxazolepropionic acid (AMPA) receptors (*Melom et al., 2013*; *Peled et al., 2014*; *Sara et al., 2011*). Synaptic *N*-methyl-*D*-aspartate (NMDA) receptors in the hippocampus also show a near complete segregation in their responses to spontaneous and evoked glutamate release (*Atasoy et al., 2008*; *Reese and Kavalali, 2016*). These studies have demonstrated that spontaneous and evoked neurotransmission can occur at the same synapse; nevertheless, these different forms of transmission activate separate NMDA and AMPA receptors. Segregation may be necessary due to the clear and distinct differences in downstream signaling between evoked and spontaneous neurotransmission at excitatory synapses (*Autry et al., 2011*;

*Crawford et al., 2017*; *Fong et al., 2015*; *Gonzalez-Islas et al., 2018*; *Nosyreva et al., 2013*; *Ramirez et al., 2017*; *Sutton et al., 2006*; *Sutton et al., 2007*; *Sutton et al., 2004*). It remains unclear whether spontaneous and evoked neurotransmission play separate roles at inhibitory synapses. In contrast to excitatory neurotransmission, inhibitory neurotransmission serves distinct functions in circuits, targets different neuronal sites, and partly relies on separate presynaptic and postsynaptic molecular machineries (*Courtney et al., 2018*; *Higley, 2014*; *Tyagarajan and Fritschy, 2014*; *Williams and Smith, 2018*). Therefore, it is critical to address directly whether receptors that respond to spontaneous and evoked GABA release are segregated at inhibitory synapses.

Here, we examine whether inhibitory synapses exhibit postsynaptic segregation of spontaneous and evoked neurotransmission. These experiments require a use-dependent γ-aminobutyric acid-A receptor (GABA$_A$R) antagonist to separate inhibitory spontaneous and evoked neurotransmission. Picrotoxin (PTX) is a commonly used noncompetitive GABA$_A$R antagonist (*Akaike et al., 1985*; *Gallagher et al., 1978*; *Nicoll and Wojtowicz, 1980*; *Takeuchi and Takeuchi, 1969*). Structural evidence suggests PTX binds within the pore of the GABA$_A$R (*Chen et al., 2006*; *Gurley et al., 1995*; *Hibbs and Gouaux, 2011*; *Masiulis et al., 2019*; *Olsen, 2006*), and may act in a use-dependent manner to block GABA$_A$R channels (*Akaike et al., 1985*; *Newland and Cull-Candy, 1992*; *Yoon et al., 1993*). However, previous studies used exogenously applied GABA to examine the pharmacology of PTX; this setting may not be completely relevant to physiological signaling mediated by synaptically released GABA. We first show PTX acts as a use-dependent GABA$_A$R antagonist during inhibitory neurotransmission. We subsequently used PTX to investigate postsynaptic segregation of spontaneous and evoked signaling at inhibitory synapses and identified partially segregated populations of GABA$_A$Rs that are solely activated by evoked release. Collectively, these results provide new insight into fundamental aspects of GABAergic neurotransmission.

## Results

### Picrotoxin acts as a use-dependent antagonist in a manner consistent with open channel block

We first tested whether PTX acts as a use-dependent GABA$_A$R antagonist to selectively block open GABA$_A$Rs activated during inhibitory neurotransmission in dissociated hippocampal cultures. Typically, blockers that gain access to their binding sites via open channel pores prematurely hinder channel conductance leading to faster decay times. Therefore, in these experiments, we examined decay times after PTX treatment. Both spontaneous and evoked currents had faster decay times after PTX (*Figure 1A–D*). Here and all subsequent experiments, we used 50 µM of PTX as this concentration of PTX was able to completely abolish spontaneous miniature inhibitory postsynaptic currents (mIPSCs) within 5 min of application (*Figure 1—figure supplement 1*, also see below). Under the same conditions, stimulating hippocampal synapses at a variety of frequencies and measuring the evoked IPSC (eIPSC) peak amplitudes in the presence of PTX resulted in a response that decreased as a function of stimulation number regardless of PTX incubation time, indicating PTX is a use-dependent antagonist (*Figure 1E*). In contrast, stimulation in the absence of PTX led to a small decrease in eIPSC peak amplitude which was far less than the decrease in the presence of PTX, and may reflect metabolic rundown. Together, these data support PTX as use-dependent antagonist during inhibitory neurotransmission.

### Kinetics of picrotoxin block correlate with presynaptic release probability at inhibitory synapses

If picrotoxin is truly a use-dependent antagonist, then the rate of GABA$_A$R block should be proportional to presynaptic release probability (as with MK801 and NMDA receptors; *Hessler et al., 1993*; *Rosenmund et al., 1993*). To test this premise, we manipulated presynaptic release probability by altering external concentrations of Ca$^{2+}$ in our dissociated culture system. Increasing Ca$^{2+}$ concentration (from 0.5 mM – 8 mM) led to an increase in the initial eIPSC response to evoked stimulation (*Figure 2A–B*). Moreover, increasing Ca$^{2+}$ concentration led to a decrease in paired pulse ratio and switched synapses from facilitation to depression, which is consistent with an increase in release probability (*Figure 2A,C*). We then stimulated neurons in the presence of PTX and recorded the progression of the eIPSC responses over 100 stimulations. Increasing release probability, via increasing

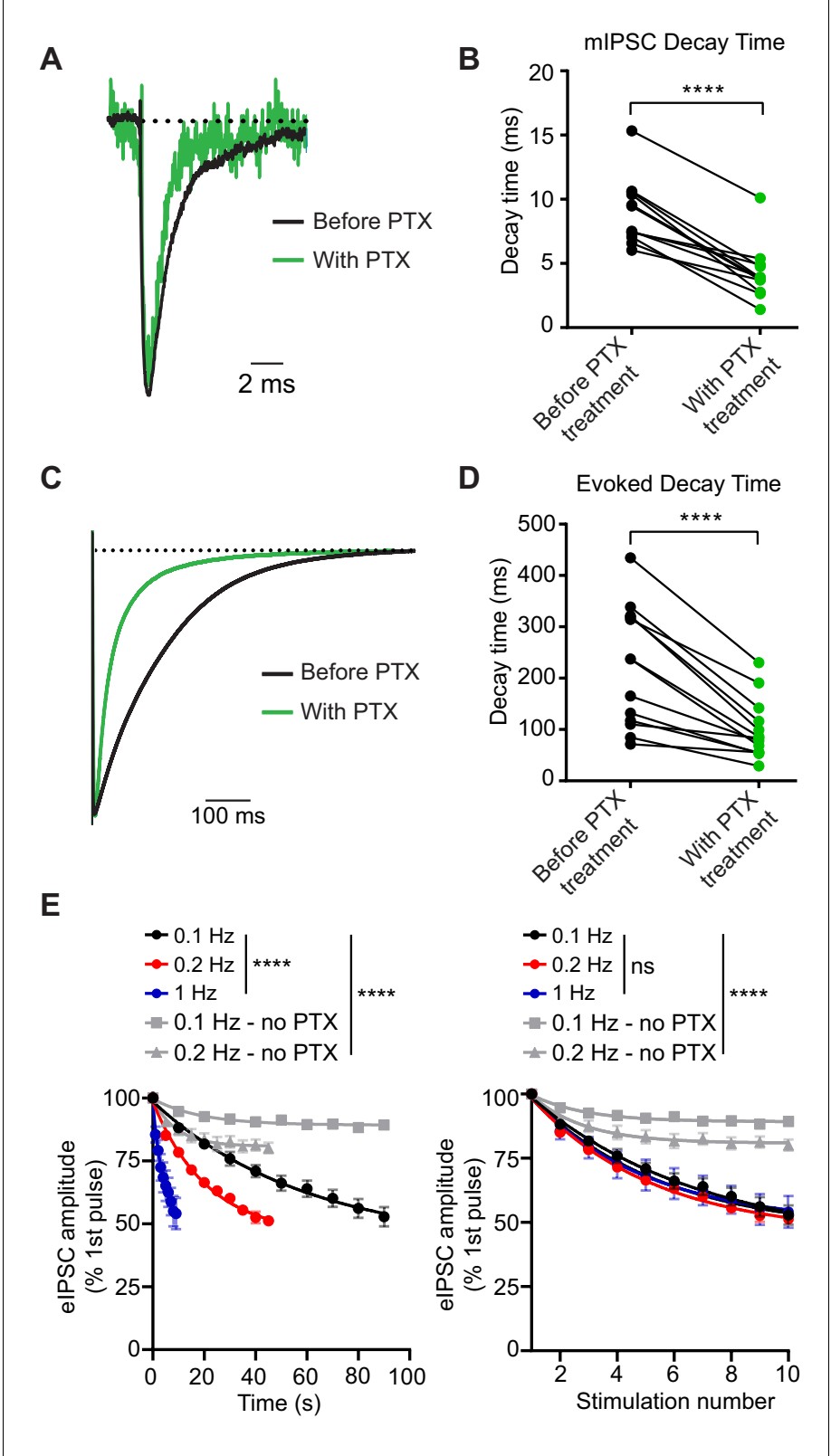

**Figure 1.** PTX blocks GABA$_A$Rs in a use-dependent manner consistent with open-channel block. (**A**) Scaled example traces of mIPSCs before and after PTX addition. (**B**) Quantification showing average event decay times obtained from the same cell before and after (8 min) treatment with 50 μM PTX. mIPSC decay time is decreased following PTX treatment (paired t-test $t_{(11)}$ = 9.055, p<0.0001, n = 12). (**C**) Scaled example traces of evoked

*Figure 1 continued on next page*

*Figure 1 continued*

responses to 0.1 Hz stimulation before and after PTX addition. Average trace taken from the 10th response to stimulation in PTX following 8 min of PTX application at rest (no stimulation). (**D**) Quantification showing average evoked response decay time obtained from the same cell before and after (8 min) treatment with 50 µM PTX. Evoked response decay time is decreased following PTX treatment (paired t-test $t_{(12)}$ = 6.097, p<0.0001, n = 13). (**E**) (Left) PTX block of evoked response plotted by total treatment time. (Right) PTX block of evoked response plotted by stimulation number. PTX blocks evoked response as a function of stimulation number, rather than time, indicating it is a use-dependent blocker (non-linear regression single exponential fit for conditions with PTX; Time: Sum-of-Squares F test $F_{(6,\ 141)}$=38.16, p<0.0001 that is one curve does not fit all datasets; Stimulation number: Sum-of-Squares F test $F_{(6,\ 141)}$=1.005, p=0.4243 that is one curve does fit all datasets, n = 5 all groups). Decay of the eIPSC response without PTX is significantly less than with PTX, indicating that rundown is not responsible for the decrease in response. (non-linear regression single exponential fit for all conditions; Time: Sum-of-Squares F test $F_{(12,\ 705)}$=115.9, p<0.0001 that is one curve does not fit all datasets; Stimulation number: Sum-of-Squares F test $F_{(12,\ 705)}$=101.9, p<0.0001 that is one curve does not fit all datasets, 0.1 Hz - no PTX n = 46, 0.2 Hz - no PTX n = 11) Graphs are mean ± SEM. **** indicates p<0.0001.

The online version of this article includes the following source data and figure supplement(s) for figure 1:

**Source data 1.** Source data for *Figure 1*.
**Figure supplement 1.** Examination of different doses of PTX.
**Figure supplement 1—source data 1.** Source data for *Figure 1—figure supplement 1*.

external $Ca^{2+}$ concentration, reliably led to a more rapid block of the eIPSC response (*Figure 2D*). These data demonstrate that PTX acts as a use-dependent blocker and can be used to monitor alterations in presynaptic release probability at inhibitory synapses.

## GABA$_A$Rs activated by spontaneous and evoked signaling show partial overlap

Our results indicate that PTX will only block GABA$_A$Rs which have been activated. Taking advantage of the use-dependency of PTX, we designed a series of experiments in dissociated hippocampal cultures to evaluate the postsynaptic segregation of inhibitory evoked and spontaneous neurotransmission. Initially, we monitored a baseline of responses to evoked stimulation, then blocked all receptors activated by spontaneous neurotransmission with PTX incubation at rest and measured the remaining evoked response (*Figure 3A*). When measuring the time course of mIPSC block in PTX, we found that 5 min of PTX treatment in the absence of stimulation was sufficient to fully block the mIPSCs (*Figure 3B–C*). Both the frequency and amplitude distributions of these spontaneous events were unaltered by the addition of tetrodotoxin (TTX), indicating that suppression of excitatory synaptic transmission to isolate inhibitory neurotransmission is in itself sufficient to block all network activity and enable detection of mIPSCs without a requirement for TTX application (*Figure 3—figure supplement 1*). It is possible that some receptors which are activated by spontaneous release remain unblocked after 5 min, but these mIPSCs may be undetectable due to a reduced size. To test this possibility, we used a high Cl⁻ internal solution to increase mIPSC amplitudes (*Figure 3D–E*). Augmented mIPSCs recorded using the high Cl⁻ internal solution followed a similar time course of block in the presence of PTX as those recorded using the standard internal solution (*Figure 3B–C*), indicating that GABA$_A$Rs which are activated by spontaneous signaling have reached a steady state of block following a 5 min incubation in PTX. Because PTX is use-dependent, and nearly all spontaneous responses are suppressed after a 5 min incubation (*Figure 3B*), the majority of remaining response to evoked stimulation after 5 min must be mediated by GABA$_A$Rs which are only responsive to evoked neurotransmission.

To evaluate whether suppression of mIPSCs also hinders subsequent evoked responses, we first established the upper and lower limits of evoked GABAergic responses to stimulation in our system. When no drug is applied during rest, the evoked response is diminished compared to the initial response before treatment (*Figure 3F*, open symbols). This may be due to metabolic rundown, as in these recordings we did not detect any alterations in membrane or pipette access resistances. Spontaneous mIPSCs, in contrast, were largely unaffected by this rundown (*Figure 3—figure supplement 1*). These data establish the upper bound of the GABA$_A$R-mediated response. To establish a lower bound for GABA$_A$R-mediated response, bicuculline was applied for 8 min. As a competitive

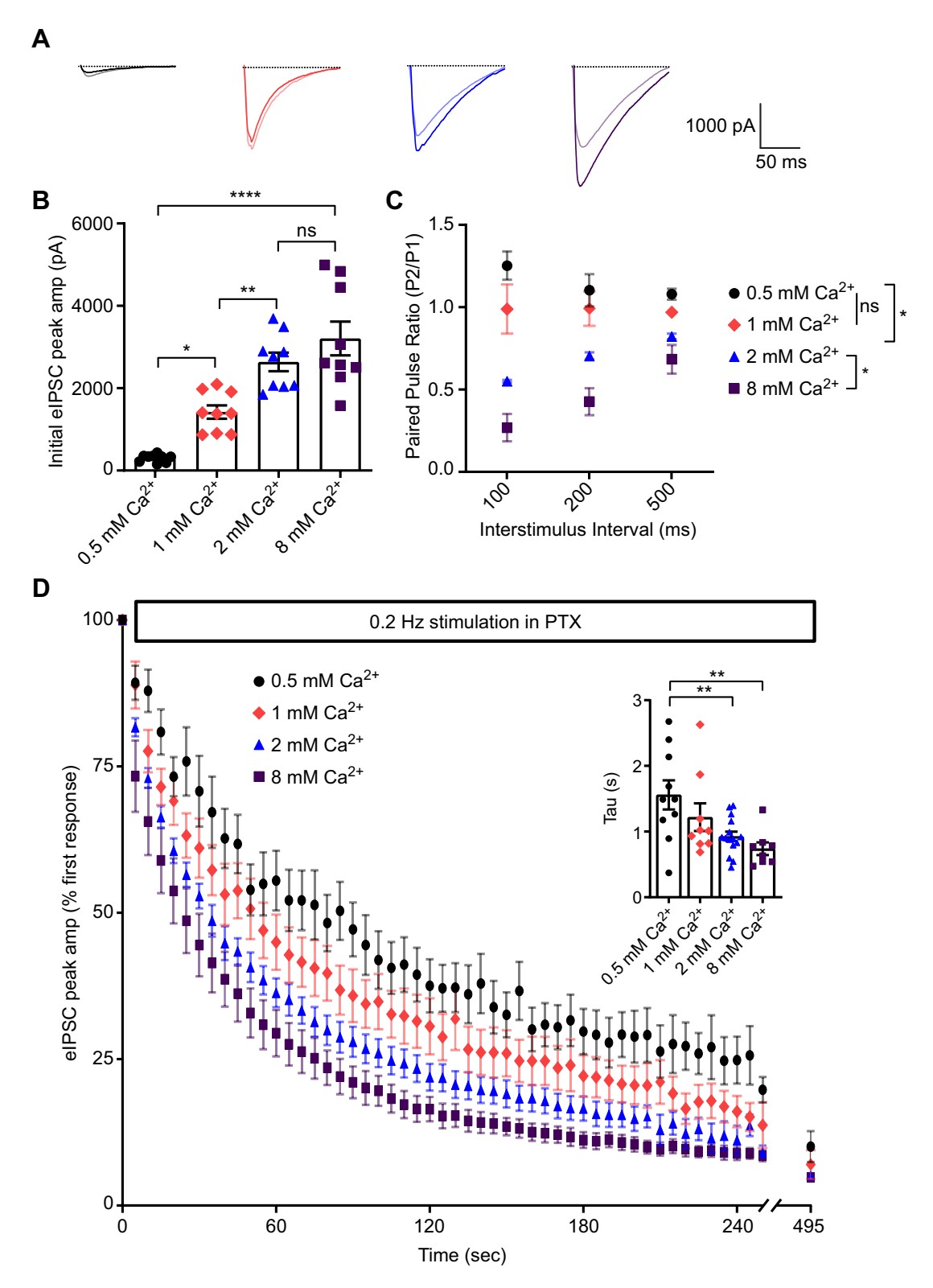

**Figure 2.** PTX can be used to compare release probability of inhibitory synapses. (**A**) Example traces of paired pulse responses at an interstimulus interval of 100 ms in 0.5 mM $Ca^{2+}$ (pulse one black, pulse two gray), 1 mM $Ca^{2+}$ (pulse one pink, pulse two light pink), 2 mM $Ca^{2+}$ (pulse one blue, pulse two light blue) or 8 mM $Ca^{2+}$ (pulse one purple, pulse two light purple). (**B**) Quantification of initial peak amplitude of eIPSC in different $Ca^{2+}$ concentrations. Increasing $Ca^{2+}$ concentration increases the initial peak eIPSC amplitude, consistent with increased release probability (one-way

*Figure 2 continued on next page*

*Figure 2 continued*

ANOVA $F_{(3,32)}$ = 27.24, p<0.0001, Tukey's post-hoc testing 0.5 mM $Ca^{2+}$ vs 1 mM $Ca^{2+}$ p=0.0160, 1 mM $Ca^{2+}$ vs 2 mM $Ca^{2+}$ p=0.0079, 2 mM $Ca^{2+}$ vs 8 mM $Ca^{2+}$ p=0.3821, 0.5 mM $Ca^{2+}$ vs 8 mM $Ca^{2+}$ p<0.0001, n = 9 all groups). (C) Paired pulse ratio recorded from cells in different $Ca^{2+}$ concentrations. Increasing $Ca^{2+}$ concentration decreased paired pulse ratio, consistent with increased release probability (two-way ANOVA interaction $F_{(6,26)}$ = 2.801, p=0.0308, interevent interval factor $F_{(2,26)}$ = 2.220, p=0.1287, $Ca^{2+}$ concentration factor $F_{(3,26)}$ = 43.55, p<0.0001, Tukey's post-hoc testing 0.5 mM $Ca^{2+}$ vs 1 mM $Ca^{2+}$ p=0.1202, 1 mM $Ca^{2+}$ vs 2 mM $Ca^{2+}$ p=0.0024, 2 mM $Ca^{2+}$ vs 8 mM $Ca^{2+}$ p=0.0107, 0.5 mM $Ca^{2+}$ vs 8 mM $Ca^{2+}$ p<0.0001, n = 3 for 0.5 mM $Ca^{2+}$; n = 3 for 1 mM $Ca^{2+}$; n = 3 for 2 mM $Ca^{2+}$; n = 4 for 8 mM $Ca^{2+}$). (D) eIPSC peak amplitude over successive 0.2 Hz stimulations in the presence of PTX. Increasing $Ca^{2+}$ concentration increased the rate of eIPSC block. (Inset) Individual time constants of single exponentials fitted to each experiment. Increasing $Ca^{2+}$ concentration decreased the time constant, consistent with an increased rate of block, demonstrating the utility of PTX to estimate release probability (one-way ANOVA $F_{(3,38)}$ = 5.125, p=0.0045, Tukey's post-hoc testing 0.5 mM $Ca^{2+}$ vs 1 mM $Ca^{2+}$ p=0.4468, 0.5 mM $Ca^{2+}$ vs 2 mM $Ca^{2+}$ p=0.0162, 0.5 mM $Ca^{2+}$ vs 8 mM $Ca^{2+}$ p=0.0062, 2 mM $Ca^{2+}$ vs 8 mM $Ca^{2+}$ p=0.8203, n=10 for 0.5 mM $Ca^{2+}$; n = 8 for 1 mM $Ca^{2+}$; n = 14 for 2 mM $Ca^{2+}$; n = 9 for 8 mM $Ca^{2+}$). Graphs are mean ± SEM. * indicates p<0.05, ** indicates p<0.01, **** indicates p<0.0001, ns indicates not significant.

The online version of this article includes the following source data for figure 2:

**Source data 1.** Source data for *Figure 2*.

antagonist, bicuculline blocks GABA_ARs regardless of whether they have been activated (*Akaike et al., 1985*; *Masiulis et al., 2019*). Response amplitudes after bicuculline incubation were greatly diminished (*Figure 3F*, purple), however, some current remained (~7%), which was sensitive to tetrodotoxin application, indicating that it was not an artifact of stimulation (*Figure 3F*, brown). This is consistent with previous studies, in which PTX achieved an imperfect block of GABA-induced current (*Akaike et al., 1985*; *Newland and Cull-Candy, 1992*; *Yoon et al., 1993*). Additionally, application of bicuculline to control drug-free conditions led to a drastic and immediate decrease in eIPSC amplitudes, indicating that responses above this baseline level represent currents mediated by GABA_ARs.

Next, we applied PTX in the absence of stimulation for 5 min, and then stimulated cells and measured the eIPSC response. If there is complete overlap between the receptors which are activated by spontaneous release and those activated by evoked release, after a 5 min incubation with PTX, when the majority of receptors activated by spontaneous release are blocked, we would expect to see no eIPSC response to stimulation above the level reached after bicuculline block. However, if the two populations of receptors are completely separate, we would expect to see a high eIPSC response comparable to the drug-free condition response. In these experiments, we found an intermediate initial eIPSC response following complete block of receptors activated by spontaneous release (*Figures 3F–H*, 5 minutes). The response decreased over successive stimulations due to the continued presence of PTX, indicating that receptors which were activated by previous evoked release are subsequently blocked. Using the initial responses in the drug-free condition and those remaining after bicuculline treatment as the maximum and minimum of the detectable GABA_AR-mediated response, we were able to calculate that 40.1 ± 9.6% of the evoked response remains after complete suppression of mIPSCs. The magnitude of the remaining evoked response was remarkably similar if the cells were incubated at rest with PTX for 8 min (39.7 ± 4.6%, *Figure 3H*). This result indicates that approximately 40% of the evoked inhibitory response is mediated by postsynaptic GABA_ARs which are exclusively activated by evoked neurotransmission, while the remaining 60% of the response is mediated by receptors which are activated by both spontaneous and evoked neurotransmission in hippocampal cultures. Consistent with a partial overlap of receptors activated by evoked and spontaneous neurotransmission, the initial response to evoked stimulation was much higher when receptors activated by spontaneous release were not fully blocked (*Figures 3F–H* and *1–2* minutes).

## Use dependence of PTX block of GABAergic transmission in hippocampal slices

To probe the postsynaptic organization of spontaneous and evoked neurotransmission within an intact synaptic circuit, we utilized ex vivo hippocampal slices from mature rats (11–13 weeks). We confirmed PTX's use-dependency in hippocampal slice by measuring evoked field Inhibitory Postsynaptic Potentials (fIPSPs) within the CA1 region in response to varying concentrations of external $Ca^{2+}$ (*Figure 4A–E*). Increasing extracellular $Ca^{2+}$ concentration between 0.5 mM, 1 mM, and 2 mM

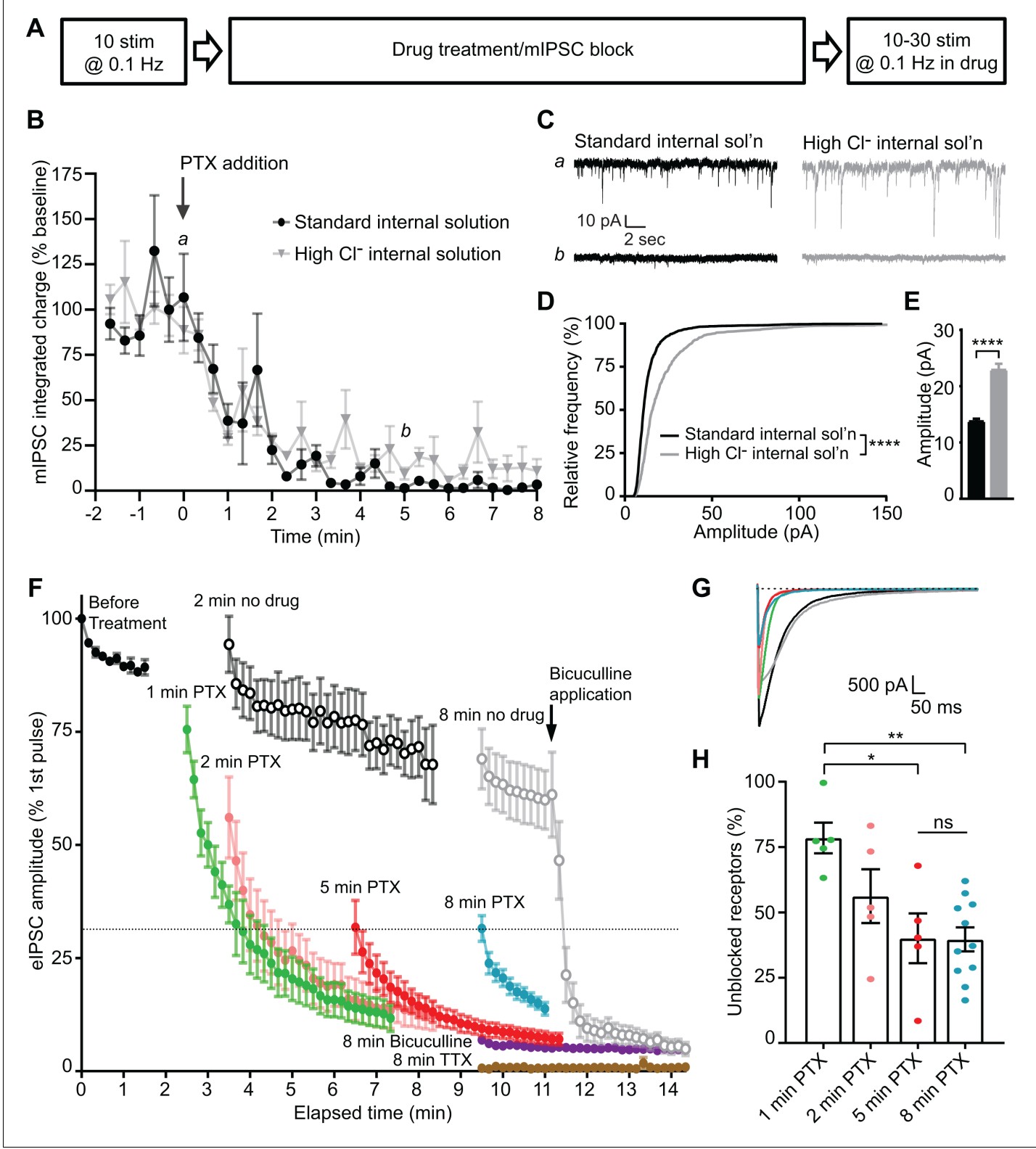

**Figure 3.** Evoked and spontaneous neurotransmission are partly segregated at inhibitory synapses. (A) Schematic showing experiment design. (B) Time course indicating mIPSC block following the addition of PTX measured using standard internal solution (black) or high Cl⁻ internal solution (gray). Integrated charge is binned in 20 s intervals. PTX diminished mIPSC frequency within 5 min. This time course is unchanged when measured using a high Cl⁻ internal solution. (C) Example traces of mIPSC recordings from indicated time points in B. (D) Cumulative histogram of spontaneous event amplitudes in standard and high Cl⁻ internal solutions. High Cl⁻ internal solution shifted the distribution of mIPSCs toward higher amplitudes

*Figure 3 continued on next page*

*Figure 3 continued*

(Kolmogrov-Smirnov test $D$ = 0.3350, p<0.0001, n = 1200 events from 12 standard internal solution recordings and 600 events from six high Cl⁻ internal solution recordings, 100 events randomly selected per recording). (E) Average of spontaneous event amplitudes in standard and high Cl⁻ internal solutions. High Cl⁻ internal solution increased the average amplitude of mIPSC events (unpaired t-test $t_{(1798)}$ = 10.96, p<0.0001, n = 1200 events from 12 standard internal solution recordings and 600 events from six high Cl⁻ internal solution recordings, 100 events randomly selected per recording). (F) Evoked inhibitory response to stimulation before drug treatment and following: no drug (open symbols, n = 6 for 2 min, n = 7 for 8 min), 1–8 min PTX (n = 5 for 1 min, n = 5 for 2 min, n = 5 for 5 min, n = 11 for 8 min), 8 min bicuculline (n = 4), or 8 min TTX treatment (n = 3). Treatment of the 8 min no drug condition with bicuculline after the 10th stimulation drastically reduced the response amplitude down to the level of 8 min bicuculline treatment, indicating that the measured response is mediated through GABA$_A$Rs. Treatment with PTX for increasing amounts of time decreased the initial evoked response to stimulation, which continued to decay upon successive stimulations in all cases. However, initial evoked response was not further decreased after a 5 min treatment with PTX. (G) Example traces of initial evoked response after PTX treatment or rest (black = 2 min no PTX, gray = 8 min no PTX, green = 1 min PTX, pink = 2 min PTX, red = 5 min PTX, blue = 8 min PTX). (H) Quantification of the percent of the initial evoked response that is mediated by GABA$_A$Rs which are unblocked following PTX treatment. Values are adjusted for bicuculline baseline and no drug treatment maximum response. After 5 min in PTX, when all receptors activated by mIPSCs are blocked, the unblocked evoked response is 40.1 ± 9.6% of the maximum response. This response is not further decreased following an 8 min treatment with PTX (39.7 ± 4.6%; one-way ANOVA $F$(3,22) = 6.228, p=0.0032, Tukey's post-hoc testing 1 min vs 2 minutes $p$=0.2260, 1 min vs 5 minutes $p$=0.0124, 1 min vs 8 minutes $p$=0.0028, 5 min vs 8 minutes $p$>0.9999, n = 5 for 1 min, n = 5 for 2 min, n = 5 for 5 min, n = 11 for 8 min). Graphs are mean ± SEM. * indicates p<0.05, ** indicates p<0.01, **** indicates p<0.0001, ns indicates not significant.

The online version of this article includes the following source data and figure supplement(s) for figure 3:

**Source data 1.** Source data for *Figure 3*.

**Figure supplement 1.** Detected spontaneous events are unaffected by TTX application.

**Figure supplement 1—source data 1.** Source data for *Figure 3—figure supplement 1*.

caused an increase in release probability, as indicated by a decrease in paired pulse ratio (*Figure 4B–C*) and increase in initial peak amplitude of fIPSPs (*Figure 4B,D*). In the presence of PTX to block GABA$_A$Rs, greater release probability caused an increase in the rate of GABA$_A$R block as evidenced by a faster and more pronounced decline in peak amplitude of fIPSPs (*Figure 4E*). These data support our earlier conclusions by demonstrating that PTX is use-dependent and can be used to compare release probability across different conditions in hippocampal slice recordings.

Next, we utilized hippocampal fIPSP recording to examine postsynaptic cross talk of GABA$_A$Rs activated by spontaneous and evoked release in hippocampal slices. Baseline responses were recorded for 10 min at 0.1 Hz stimulation followed by bath application of PTX at rest for 0, 5 or 10 min. We then continued to perfuse PTX and resumed 0.1 Hz stimulation for 30–40 min. (*Figure 4F*). Our previous data show that 5 min of PTX application at rest is sufficient to block spontaneous activity in cultured hippocampal neurons, and it has been reported that the rates of spontaneous neurotransmission are similar between cultured neurons and brain slices (*Ertunc et al., 2007*; *Kavalali, 2015*; *Sara et al., 2005*). Therefore, we measured the remaining evoked response in 5 min intervals following PTX application at rest, as the time for PTX to perfuse into the recording chamber and reach full concentration may differ between our culture and slice recordings. Accordingly, multiple minutes passed before the fIPSP response was reduced following the start of PTX perfusion when stimulation was continuously given (*Figure 4E,G*). Additionally, fIPSP peak amplitude was reduced compared to baseline when stimulation was resumed at 5 and 10 min, but the extent of block was much higher following 10 min of application at rest (*Figure 4G–I*). The decreased response following PTX application in the absence of stimulation indicates a high degree of overlap between GABA$_A$Rs activated by evoked and spontaneous neurotransmission. Conversely, following 10 min of PTX application at rest, when most receptors activated by spontaneous signaling are expected to be suppressed, there is a remaining evoked response, suggesting that this response is mediated by GABA$_A$Rs activated specifically by evoked stimulation consistent with our prior findings. Remarkably, this remaining response was similar in magnitude to our results in culture (34.05 ± 5.37%).

## Recovery of spontaneous neurotransmission following GABA$_A$R block

Our findings so far suggest that evoked and spontaneous neurotransmission partially overlap and that a population of GABA$_A$Rs are solely activated by evoked release by examining the remaining evoked responses following GABA$_A$R block during rest. To investigate this further, we examined the

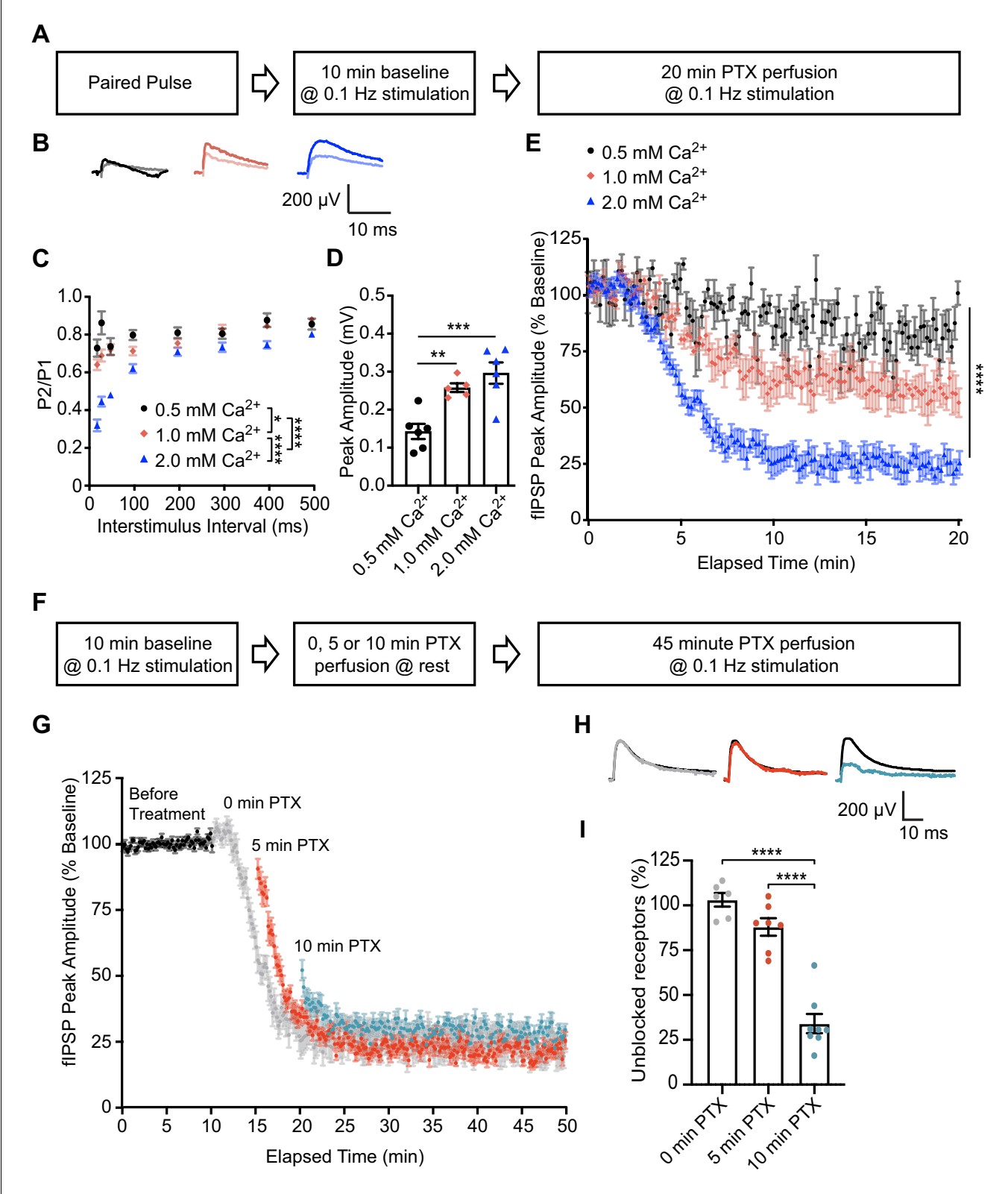

**Figure 4.** PTX exhibits use-dependency in hippocampal slices and demonstrates partial segregation of evoked and spontaneous neurotransmission at inhibitory synapses. (**A**) Schematic showing experimental design in B-E. (**B**) Averaged fIPSP paired pulse representative traces at an interstimulus interval of 100 ms in 0.5 mM $Ca^{2+}$ (pulse one black, pulse two gray), 1 mM $Ca^{2+}$ (pulse one pink, pulse two light pink) or 2 mM $Ca^{2+}$ (pulse one blue, pulse two light blue). (**C**) Paired pulse ratio (PPR) (P2/P1) was lower in 2 mM extracellular $Ca^{2+}$ than in 0.5 mM $Ca^{2+}$ or 1 mM $Ca^{2+}$ and lower in 1 mM $Ca^{2+}$ than

*Figure 4 continued on next page*

Figure 4 continued

0.5 mM Ca$^{2+}$, indicating that extracellular Ca$^{2+}$ concentration is positively associated with presynaptic release probability (repeated measures two-way ANOVA $F_{(2,48)}$ = 45.96, p<0.0001, Tukey's post hoc testing 0.5 mM Ca$^{2+}$ vs 1 mM Ca$^{2+}$ p=0.0145, 0.5 mM Ca$^{2+}$ vs 2 mM Ca$^{2+}$ p<0.0001, 1 mM Ca$^{2+}$ vs 2 mM Ca$^{2+}$ p<0.0001, n = 6 for 0.5 mM Ca$^{2+}$, n = 5 for 1 mM Ca$^{2+}$, n = 6 for 2 mM Ca$^{2+}$). (D) Quantification of baseline peak amplitudes confirming that greater extracellular Ca$^{2+}$ concentration increases presynaptic release probability and is associated with greater peak amplitude of fIPSPs (one-way ANOVA $F_{(2,14)}$ = 13.85, p=0.0005, Tukey's post-hoc testing 0.5 mM Ca$^{2+}$ vs 1 mM Ca$^{2+}$ p=0.0072, 0.5 mM Ca$^{2+}$ vs 2 mM Ca$^{2+}$ p=0.0005, n = 6 for 0.5 mM Ca$^{2+}$, n = 5 for 1 mM Ca$^{2+}$, n = 6 for 2 mM Ca$^{2+}$). (E) Time course showing block of 0.1 Hz evoked fIPSPs following PTX application in 0.5 mM Ca$^{2+}$ (black), 1 mM Ca$^{2+}$ (pink) or 2 mM Ca$^{2+}$ (blue). Greater presynaptic release probability via increased extracellular Ca$^{2+}$ is associated with faster block of GABA$_A$Rs (non-linear regression single exponential fit, Sum-of-Squares F test $F_{(6, 2048)}$=571.3, p<0.0001, n = 6 for 0.5 mM Ca$^{2+}$, n = 5 for 1 mM Ca$^{2+}$, n = 6 for 2 mM Ca$^{2+}$). (F) Schematic showing experimental design in G-I. (G) Time course showing evoked fIPSP response to 0.1 Hz stimulation before (black) and following application of PTX at rest for 0 min (gray), 5 min (red) or 10 min (blue). Treatment of PTX for increasing amounts of time resulted in a lesser remaining response upon continuation of 0.1 Hz stimulation (n = 6 for 0 min, n = 7 for 5 min, n = 8 for 10 min). (H) Averaged fIPSP representative traces at baseline (black) and first fIPSP response following 0 min (gray), 5 min (red) or 10 min (blue) of PTX administration at rest. (I) Quantification of unblocked GABA$_A$R mediating response remaining after 0, 5 or 10 min perfusion of PTX at rest, adjusted for upper and lower boundaries of fIPSP response. Following PTX administration at rest, the unblocked evoked response is 103.12 ± 3.82% of baseline after 0 min (gray), 88.0 ± 4.89% of baseline after 5 min (red) and 34.05 ± 5.37% of baseline after 10 min (blue) (one-way ANOVA $F_{(2,18)}$ = 56.30, p<0.0001, Tukey's post-hoc testing 0 min vs 10 min p<0.0001, 5 min vs 10 min p<0.0001, n = 6 for 0 min, n = 7 for 5 min, n = 8 for 10 min). Thus a response to evoked stimulation remains following 10 min of GABA$_A$R block by spontaneous neurotransmission, indicating partial segregation. Graphs are mean ± SEM. * indicates p<0.05, ** indicates p<0.01, *** indicates p<0.001, **** indicates p<0.0001.

The online version of this article includes the following source data for figure 4:

**Source data 1.** Source data for *Figure 4*.

kinetics of recovery from GABA$_A$R block in our dissociated culture system. For this purpose, we blocked GABA$_A$Rs by applying PTX in the presence of 55 mM of KCl for 90 s then measuring the recovery of mIPSCs from PTX block in the presence or absence of stimulation (*Figure 5A*). As expected, the eIPSC response was greatly diminished when 55 mM of KCl alone was applied for 90 s, likely due to rapid depletion of vesicle pools (*Sara et al., 2002*). Nevertheless, the eIPSC responses recovered back to a steady baseline following removal of the KCl solution. eIPSC responses following 55 mM of KCl with PTX treatment also recovered following successive stimulations, however the recovery occurred over a longer time period, consistent with the premise that blocked receptors were gradually unblocked by successive stimulations (*Figure 5B–C*). This finding agrees with previous studies (*Newland and Cull-Candy, 1992*) where PTX unblock from GABA$_A$Rs has been shown to be at least partially use-dependent (*Figure 5—figure supplement 1*). We also examined mIPSC responses to 55 mM of KCl with and without PTX in the presence or absence of this stimulation. Following application of 55 mM of KCl, mIPSC responses were partly diminished, but then recovered back to a steady baseline (*Figure 5D–E*). When 55 mM of KCl was co-applied with PTX, mIPSC responses were greatly diminished, indicating GABA$_A$Rs activated by spontaneous release were blocked by 55 mM of KCl and PTX co-application. The recovery of mIPSCs in cells treated with 55 mM of KCl and PTX in the absence of stimulation was slow and minimal (*Figure 5D–E*). This is not surprising since the spontaneous release rate per synapse is in the order of 1 vesicle per minute per synapse (*Leitz and Kavalali, 2014*; *Sara et al., 2005*). However, the presence of 0.2 Hz stimulation significantly increased the recovery of mIPSCs (*Figure 5F*), indicating that evoked release activates some of the same GABA$_A$Rs as spontaneous release. To ensure our results were not being biased towards receptors which are specifically activated by evoked release caused by depolarization, we repeated this experiment using the GABA$_A$R agonist muscimol (10 µM) in combination with PTX to block GABA$_A$Rs (*Figure 6*). We obtained similar results using this approach, that is stimulation significantly increased the recovery of mIPSCs following GABA$_A$R block (*Figure 6D–E*). These results support the notion that GABA$_A$Rs which are activated by spontaneous and evoked release partially overlap, but do not exclude the existence of a fraction of GABA$_A$Rs which are only activated by spontaneous release.

## Discussion

In this study, we show that PTX blocks synaptic GABA$_A$Rs in a use-dependent manner. This use-dependent action of PTX allowed us to investigate fundamental properties of GABAergic inhibitory synapses. By altering presynaptic release probability via changes in external Ca$^{2+}$, we show that PTX

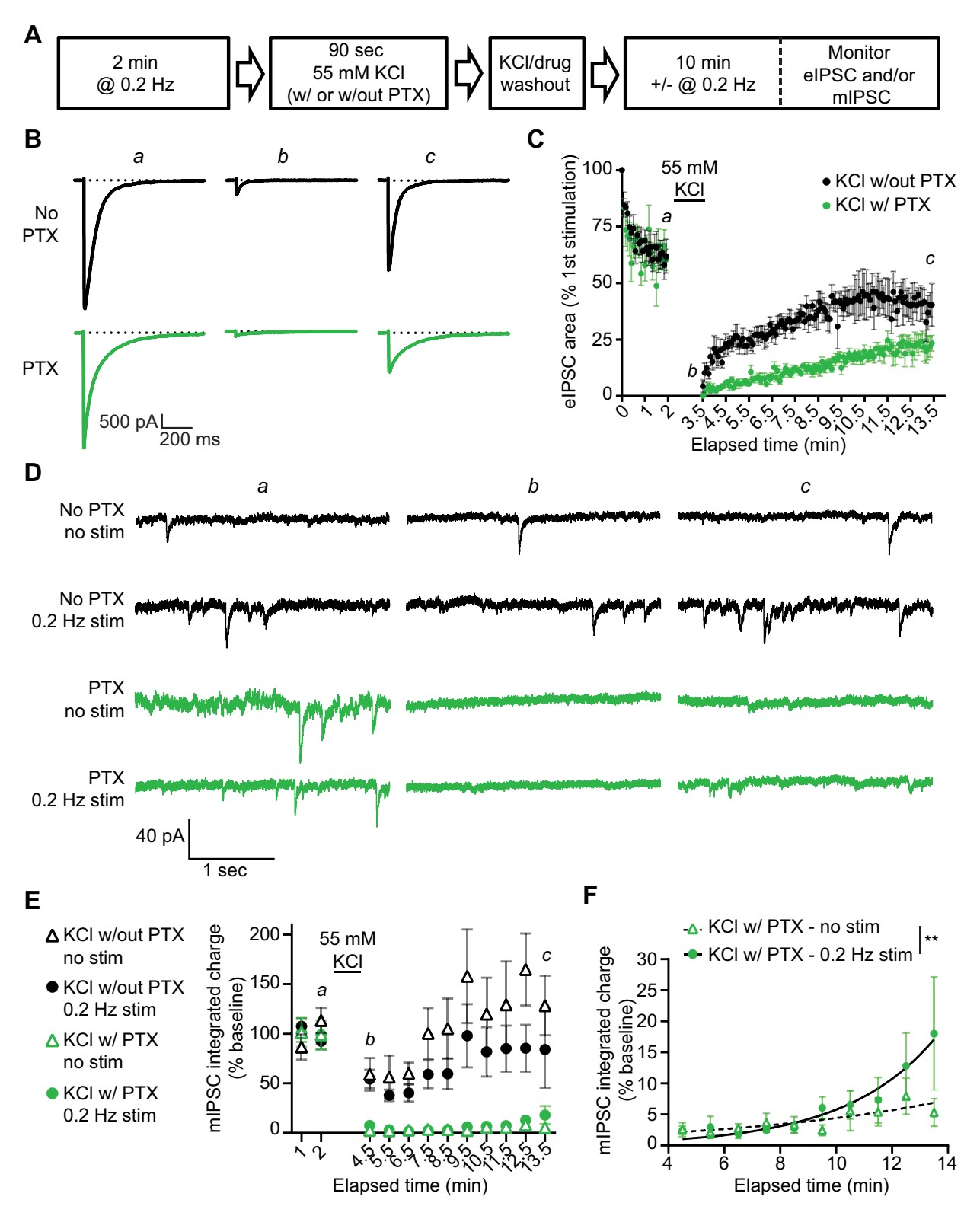

**Figure 5.** Recovery from PTX block of receptors activated by spontaneous GABA release is enhanced by the administration of evoked stimulation. (**A**) Schematic showing experiment design. (**B**) Example traces of eIPSC responses from time points indicated in C. Stimulus artifacts were removed for clarity. (**C**) Quantification of eIPSC recovery following KCl treatment with and without PTX. KCl treatment initially depresses the eIPSC response. The evoked response is recovered after KCl treatment without PTX, but recovery after KCl treatment with PTX proceeds over a longer time course

*Figure 5 continued on next page*

*Figure 5 continued*

indicating GABA$_A$Rs which were activated by KCl treatment and blocked by PTX are being unblocked with successive stimulations. (**D**) Example traces of mIPSCs from the timepoints indicated in E recorded in the presence or absence of stimulation before and after KCl treatment with and without PTX. (**E**) Quantification of mIPSC recovery (1 min bins) in the presence or absence of stimulation following KCl treatment with and without PTX. KCl treatment without PTX initially depresses mIPSCs, but does not fully block them. KCl treatment with PTX blocks mIPSCs, after which they recover slowly (KCl w/out PTX – no stim, n = 6; KCl w/out PTX – 0.2 Hz stim, n = 5; KCl w/PTX – no stim, n = 6; KCl w/PTX – 0.2 Hz stim, n = 6). (**F**) Analysis of mIPSC recovery (1 min bins) in PTX treated samples in the absence (open triangles) and presence (closed circles) of 0.2 Hz stimulation. Data were fitted with non-linear regression exponential model (no stimulation, dashed curve, n = 6; 0.2 Hz stimulation, solid curve, n = 6) which were significantly different from each other (Sum-of-squares F test $F_{(2,109)}$ = 6.976, p=0.0014). These data indicate that stimulation increases the recovery of mIPSCs, consistent with partial overlap of receptors activated by evoked and spontaneous signaling at inhibitory synapses. Graphs are mean ± SEM. ** indicates p<0.01.

The online version of this article includes the following source data and figure supplement(s) for figure 5:

**Source data 1.** Source data for *Figure 5*.
**Figure supplement 1.** Rate of unblock following PTX application is use-dependent.
**Figure supplement 1—source data 1.** Source data for *Figure 5—figure supplement 1*.

can be used to interrogate release probability at inhibitory synapses in both cultures and hippocampal slices. Additionally, we were able to test whether evoked and spontaneous neurotransmission processes are segregated at inhibitory synapses. Our data show that while it is not as complete as previously reported at excitatory synapses (*Atasoy et al., 2008*; *Melom et al., 2013*; *Peled et al., 2014*; *Reese and Kavalali, 2016*; *Sara et al., 2011*), there is nevertheless significant postsynaptic segregation of the two forms of neurotransmission at inhibitory synapses.

The use-dependent NMDA receptor blocker MK-801 has been instrumental in determining basic properties of glutamatergic synapses such as presynaptic release probability (*Hessler et al., 1993*; *Huang and Stevens, 1997*; *Huettner and Bean, 1988*; *Rosenmund et al., 1993*), receptor saturation (*McAllister and Stevens, 2000*), and receptor mobility (*Tovar and Westbrook, 2002*). Our results show that PTX is similarly use-dependent and can be used to monitor alterations in release probability of GABAergic synapses. By increasing extracellular Ca$^{2+}$ levels, we gradually augmented release probability at inhibitory synapses and observed a corresponding increase in the rate at which GABAergic responses were blocked. It is therefore likely that many of the same analyses initially performed for NMDA receptors at glutamatergic synapses can be performed for GABA$_A$Rs at GABAergic synapses. However, it is important to caution that PTX does not simply block pore conductance of GABA channels but rather binds to a site within the pore to stabilize a closed state (*Masiulis et al., 2019*). In addition, recovery from block is only partially use-dependent as indicated by earlier work (*Newland and Cull-Candy, 1992*). Although these properties of PTX have limited impact on the experiments we performed here, they may influence alternative experimental designs such as delineating metabotropic and ionic receptor effects (*Nabavi et al., 2013*), or those that solely rely on response recovery after block.

Our results show that evoked and spontaneous neurotransmission processes are partially segregated at inhibitory synapses. This notion is supported by the GABA$_A$R-mediated evoked responses remaining following the block of all GABA$_A$Rs activated by spontaneous neurotransmission in both hippocampal cultures and slice. We estimate that 34–40% of the total evoked GABAergic response recorded at the stratum pyramidale of hippocampal CA1 region and in hippocampal cultures is mediated by receptors which are only activated by evoked neurotransmission. Although mIPSC recovery following GABA$_A$R block was enhanced by the presence of stimulation, these results do not exclude the possibility that some GABA$_A$Rs are only activated by spontaneous release and account for the population that recovers in the absence of stimulation. These findings suggest a model of partial, but not complete, segregation between GABA$_A$Rs which are activated by evoked release, and those activated by spontaneous release (*Figure 6F*).

A major question arising from this work is: how is this partial segregation achieved? Segregation may start at the level of presynaptic release mechanisms, including dedicated signaling pathways that selectively impact spontaneous or evoked release as well as distinct synaptic vesicle pools and vesicle recycling pathways to carry out the two forms of neurotransmission (*Abrahamsson et al., 2017*; *Kavalali, 2015*). A growing body of evidence suggests that synapses are organized into 'nano-columns' such that a single vesicle only activates juxtaposed receptors on the postsynaptic membrane (*Biederer et al., 2017*; *Maschi and Klyachko, 2017*; *Tang et al., 2016*). Based on these

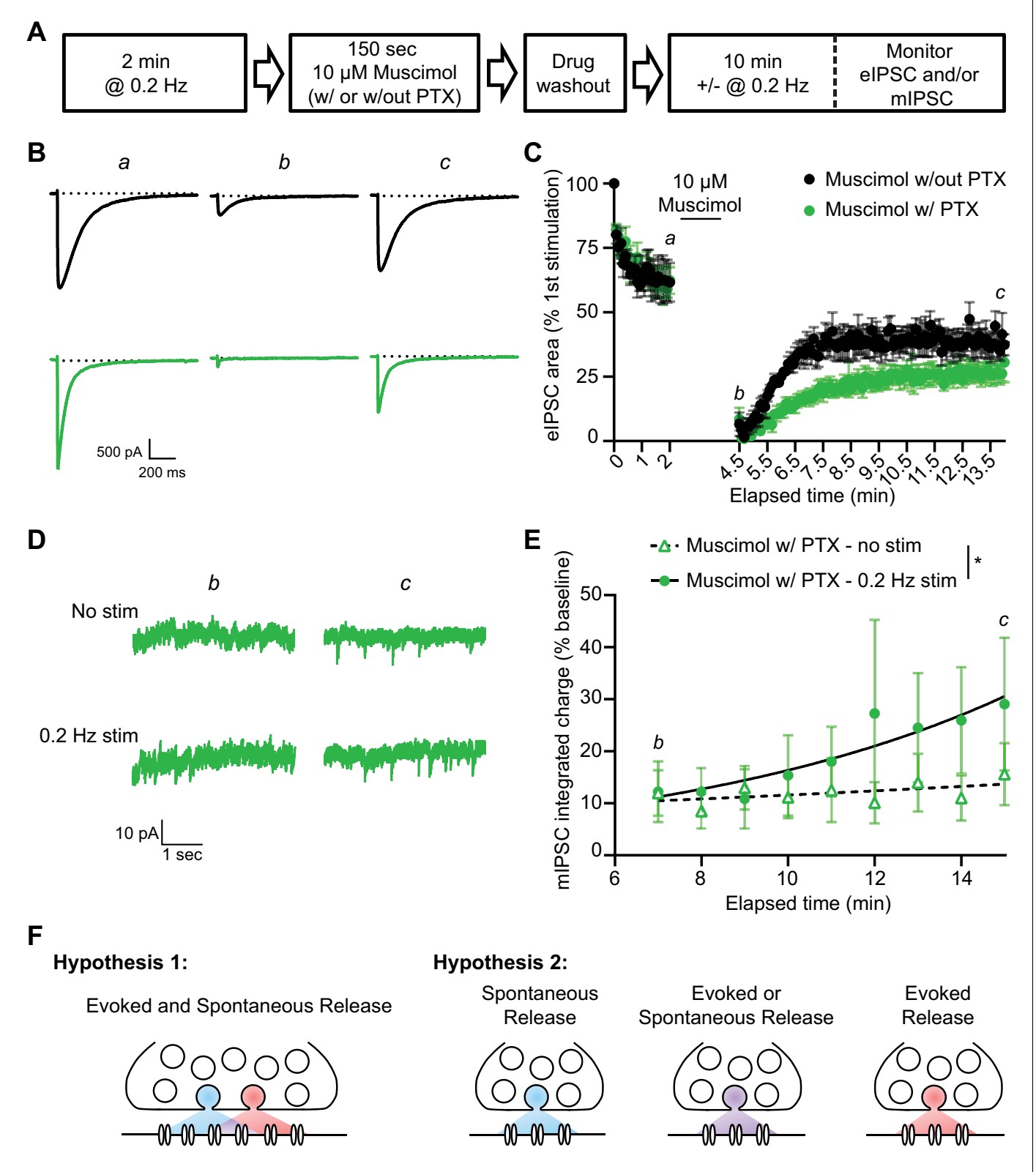

**Figure 6.** A similar pattern of mIPSC recovery from block is detected when using muscimol to activate GABA_ARs during PTX block. (**A**) Schematic showing experiment design. (**B**) Example traces of eIPSC responses from time points indicated in C. Stimulus artifacts removed for clarity. (**C**) Quantification of eIPSC recovery following muscimol treatment with and without PTX (n = 4, all groups). Muscimol treatment initially depresses the eIPSC response. The evoked response is recovered after muscimol treatment without PTX, but recovery after muscimol treatment with PTX proceeds
*Figure 6 continued on next page*

*Figure 6 continued*

over a longer time course indicating GABA$_A$Rs which were activated by muscimol treatment and blocked by PTX are being unblocked with successive stimulations. (D) Example traces of mIPSCs from the timepoints indicated in E. (E) Analysis of mIPSC recovery (1 min bins) in PTX treated samples in the absence (open triangles) and presence (closed circles) of 0.2 Hz stimulation. Data were fitted with non-linear regression exponential models (no stimulation, dashed curve, n = 6; 0.2 Hz stimulation, solid curve, n = 3) which were significantly different from each other (Sum-of-squares F test $F_{(2,74)}$ = 4.832, p=0.0107). These data indicate that stimulation increased the recovery of mIPSCs, consistent with partial overlap of receptors activated by evoked and spontaneous signaling at inhibitory synapses. (F) Graphic summary of findings indicating a partial overlap of receptors activated by spontaneous and evoked GABA release. Partial overlap could be achieved either through spatial segregation of evoked and spontaneous presynaptic release and postsynaptic receptors within the same synapse, or through specialization of synapses for either spontaneous release, evoked release, or both. Graphs are mean ± SEM. * indicates p<0.05.

The online version of this article includes the following source data for figure 6:

**Source data 1.** Source data for *Figure 6*.

studies, two non-mutually exclusive models can be proposed. One model posits that segregation is achieved through a combination of differential presynaptic molecular regulation and pre/postsynaptic spatial separation within the same synapse. Alternatively, segregation may arise through differential action of these mechanisms over different inhibitory synapses. Earlier studies demonstrated that excitatory synapses can exhibit both spontaneous and evoked release while others exclusively exhibit spontaneous or evoked release (*Leitz and Kavalali, 2014*; *Melom et al., 2013*; *Peled et al., 2014*; *Reese and Kavalali, 2016*; *Walter et al., 2014*). With GABAergic neurotransmission, limitations in the resolution and speed of current chloride sensors or other probes to visualize activity at the level of single synapses, makes this question difficult to address at this time. Therefore, we cannot exclude the possibility that some GABAergic synapses could release neurotransmitter only via spontaneous release, only via evoked release, or both leading to the partial segregation seen in our experiments (*Figure 6F*).

There could be many reasons why segregation is not as complete at inhibitory synapses as at excitatory synapses. First, there may be more overlap in the molecular machinery used for spontaneous and evoked neurotransmission at inhibitory synapses. Although earlier work has suggested that synaptic vesicle pools mediating spontaneous and evoked release may be distinct at inhibitory synapses (*Chung et al., 2010*), unlike at excitatory synapses, spontaneous release at inhibitory synapses is partly coupled to activation of voltage activated Ca$^{2+}$ channels (*Tsintsadze et al., 2017*; *Vyleta and Smith, 2011*; *Williams and Smith, 2018*). Moreover, recent evidence suggests that synaptotagmin-1, the canonical Ca$^{2+}$ sensor for fast synchronous evoked release (*Geppert et al., 1994*), also regulates Ca$^{2+}$ dependent spontaneous release, but only at inhibitory synapses (*Courtney et al., 2018*). Second, unlike Ca$^{2+}$ signaling through NMDA receptors, chloride flow through GABA$_A$Rs has very few targets that may direct downstream biochemical signaling (*Chen et al., 2019*; *Heubl et al., 2017*; *Piala et al., 2014*), although GABAergic transmission has been shown to indirectly regulate dendritic Ca$^{2+}$ signaling (*Chiu et al., 2013*; *Higley, 2014*). Therefore, the strong segregation that is seen at excitatory synapses may not be as essential to mediate inhibitory synaptic communication. Our current findings provide evidence that the postsynaptic organization of inhibitory synapses shows only partial segregation of GABA$_A$Rs that respond to evoked and spontaneous neurotransmission (*Figure 6F*), suggesting a distinct organizing principle of these synapses compared to their excitatory counterparts. Nevertheless, given recently identified targets for chloride-mediated biochemical signaling in neurons (*Chen et al., 2019*; *Heubl et al., 2017*; *Piala et al., 2014*), even this partial segregation may provide a platform for differential signaling by evoked and spontaneous GABAergic neurotransmission in the central nervous system.

# Materials and methods

## Animals

Adult female dam CD1 rats (Sprague-Dawley, Charles River) aged 11–13 weeks were housed on a 12 hr light/dark cycle at ambient temperature (23 ± 3°C and humidity (50 ± 20%) with access to food and water ad libitum. Dams were used for hippocampal slice experiments. Rat pups (postnatal day (PN) 1–3) were used for culture experiments. Animal procedures conformed to the Guide for the

Care and Use of Laboratory Animals and were approved by the Institutional Animal Care and Use Committees at UT Southwestern Medical Center and at Vanderbilt University.

## Cell culture

Dissociated hippocampal cultures were prepared as previously described (*Kavalali et al., 1999*). Briefly, whole hippocampi were dissected from PN1-3 rats, trypsinized (~4 mg/mL, Sigma), mechanically dissociated, and plated on matrigel (BD Biosciences) coated glass coverslips. Neurons were plated in MEM containing 27.8 mM of Glucose, 2.4 mM of NaHCO$_3$, 1.3 µM of Transferrin (Calbiochem), 2 mM of L-Glutamine, 4.4 µM of insulin, and 10% FBS. On days in vitro (DIV) 1, FBS concentration was reduced to 5%, L-Glutamine concentration was reduced to 500 µM, and 1x B-27 supplement (Gibco) and 4 µM of cytosine arabinoside (ARAC; Sigma) were added. On DIV4 the concentration of ARAC was reduced to 2 µM. Cells were maintained at 37°C in 5% CO$_2$ atmosphere.

## Hippocampal culture whole cell recordings

Electrophysiology was performed as previously described (*Nelson et al., 2006*). On DIV15-21 cells were recorded using the whole cell voltage clamp configuration of the patch clamp technique. Extracellular Tyrode solution, adjusted to pH 7.4 and 319 mOsm, contained: 150 mM of NaCl, 4 mM of KCl, 1.25 mM of MgCl$_2$·6H$_2$O, 10 mM of glucose, 10 mM of HEPES, 2 mM of Ca$^{2+}$ (unless otherwise noted), 50 µM of D-APV, and 10 µM of NBQX. Intracellular pipette solution, adjusted to pH 7.3 and 304 mOsm, contained: 115 mM of Cs-methanesulphonate, 10 mM of CsCl, 5 mM of NaCl, 10 mM of HEPES, 20 mM of TEA.Cl hydrate, 4 mM of MgATP, 0.3 mM of GTP, 0.6 mM of EGTA, and 10 mM of QX314. High Chloride intracellular pipette solution, adjusted to pH 7.3 and 304 mOsm, contained: 122 mM of CsCl, 9 mM of NaCl, 1.8 mM of MgCl$_2$, 9 mM of EGTA, 9 mM of HEPES, 14 mM of creatine phosphate (Tris salt), 4 mM of MgATP, and 0.3 mM of Tris-GTP. Drugs in all experiments were used at the following concentrations unless otherwise noted: 50 µM of picrotoxin (Sigma Aldrich), 20 µM of bicuculline (Sigma Aldrich), 50 µM of D-APV (Abcam), 10 µM of NBQX (Abcam), 1 µM of TTX (ENZO Life Sciences), and 10 µM of muscimol (Tocris). Cells were held at −70 mV and liquid junction potential was not corrected for. eIPSCs were elicited by bath stimulation through parallel platinum electrodes. eIPSCs were on the order of 2–4 nA. Series resistance compensation was not employed in these experiments and series resistance measures were 10 MΩ or below. Series resistance was monitored before the start and following the conclusion of every recording. In cases where multiple files were recorded from the same patch, series resistance was monitored in between each file. For mIPSC data in *Figure 1* and *Figure 1—figure supplement 1*, 'before PTX treatment' data were extracted from a 2 min spontaneous baseline recording before drug treatment, and 'with PTX treatment' data were extracted from the final 2 min of the 8 min drug treatment from the experiment in *Figure 3F*. For mIPSC data in *Figure 3—figure supplement 1*, 'before TTX' data were extracted from a 30 s spontaneous baseline recording before TTX treatment, and 'after TTX' was extracted from the final 30 s of a subsequent 10 min recording. Evoked inhibitory currents were analyzed using pCLAMP10 software (Molecular Devices). Spontaneous inhibitory currents were analyzed using MiniAnalysis software (Synaptosoft). For mIPSC time courses, integrated charge was measured as the summed area of each event within each bin.

## Hippocampal slice electrophysiology

Adult female dam rats aged 11–13 weeks were used for all hippocampal slice experiments. No more than four hippocampal slices were used per dam, and each experiment was performed on slices from at least two different animals. Dams were anesthetized with isofluorane and decapitated. Brains were removed and immersed in ice-cold dissection buffer containing the following: 2.6 mM of KCl, 1.25 mM of NaH$_2$PO$_4$, 26 mM of NaHCO$_3$, 0.5 mM of CaCl$_2$, 5 mM of MgCl$_2$, 212 mM of sucrose and 10 mM of glucose for 2–3 min. Hippocampi were dissected and cut with a vibratome into 400 µm-thick transverse sections in ice-cold dissection buffer continuously aerated with 95% O$_2$ and 5% CO$_2$. Area CA3 was surgically removed immediately after sectioning. Sections were recovered for 3–6 hr at 30°C in oxygenated artificial cerebrospinal fluid (ACSF) containing the following: 124 mM of NaCl, 5 mM of KCl, 1.25 mM of NaH$_2$PO$_4$, 26 mM of NaHCO$_3$, 2 mM of CaCl$_2$, 1 mM of MgCl$_2$ and 10 mM of glucose, pH 7.4 (continuously aerated with 95% O$_2$ and 5% CO$_2$). 25 µM of D-APV and 10 µM of NBQX were added to the ACSF to block NMDA and AMPA channels, respectively, and isolate

inhibitory GABA-mediated currents. In several experiments, PTX was applied for long durations to fully block GABA$_A$Rs. Following the conclusion of these experiments, bicuculline was applied to confirm the range of inhibitory response. Bicuculline did not further decrease the measured response, indicating that as for dissociated culture experiments, PTX blocked the full range of the GABA$_A$R-mediated response. For experiments examining release probability, concentration of CaCl$_2$ in ACSF varied at 2 mM, 1 mM and 0.5 mM. Following recovery, hippocampal slices were transferred to the recording chamber and perfused with oxygenated ACSF at a rate of 2–3 mL/min at 30°C. Field Inhibitory Postsynaptic Potentials (fIPSPs) were evoked by inserting a concentric bipolar stimulating electrode (FHC Inc) and an extracellular recording electrode filled with ACSF (resistance, 1–2 MΩ) proximally in *s. pyramidale* of CA1 just below the surface of the tissue *Younts et al., 2016*.

In all experiments, paired pulse ratio was measured by eliciting paired-pulse stimulations at decreasing interstimulus intervals (ISIs) of 500, 400, 300, 200, 100, 50, 20 and 30 ms. The fIPSP peak amplitude of pulse 2 (P2) was divided by pulse 1 (P1) to give a ratio representing presynaptic release probability. Baseline responses were then collected every 10 s (0.1 Hz) using an input stimulus intensity that induced 75% of the slice's maximum response for 10 min. All data presented are the mean ± SEM of individual trials, and data are not binned. Specific n numbers for each experiment are specified in the figure legends.

To examine the use-dependency of picrotoxin (*Figure 4A–E*), extracellular Ca$^{2+}$ in ACSF was varied between 0.5 mM, 1 mM and 2 mM to alter presynaptic release probability. 50 µM of picrotoxin was then perfused at a flow rate of 18.0 mL/hour with 0.1 Hz stimulation and fIPSP peak amplitude was recorded for 20 min to produce a decay curve of GABA$_A$R block. 75% of maximal response varied from 85 to 223 µV response elicited by a 35–40 µA stimulation for 0.5 mM Ca$^{2+}$, 231–297 µV response elicited by a 15–39 µA stimulation for 1.0 mM Ca$^{2+}$, and 175–358 µV response elicited by a 12–15 µA stimulation for 2.0 mM Ca$^{2+}$. To investigate evoked GABA$_A$R activity following block of spontaneous GABA$_A$R (*Figure 4F–I*), 50 µM of picrotoxin was perfused at a flow rate of 18.0 mL/hour at rest for 0, 5 or 10 min. 0.1 Hz stimulation was then resumed and picrotoxin perfusion continued for 40 min. 75% of maximal response varied from 175 to 358 µV response elicited by a 12–15 µA for 0 min, 208–361 µV response elicited by a 9–25 µA stimulation for 5 min, and 245–460 µV response elicited by a 11–18 µA stimulation for 10 min.

Data were analyzed by normalizing peak response amplitudes to average baseline peak amplitude for each recording. Unblocked receptor percentage following 0, 5 or 10 min PTX perfusion at rest was quantified by determining upper and lower limits of fIPSP response using the average of the first and last 50 stimulations of each recording, respectively. The lower limit was then subtracted from the first fIPSP response following the continuation of stimulation, and this value was divided by the range and subtracted from one to determine the percentage of baseline response that was intact. Slices were visually inspected before experiments and continually monitored throughout. Recordings with unstable initial baselines or presynaptic fiber volleys were excluded from analysis.

## Statistical analyses

All statistical analyses were performed using GraphPad Prism 7 or 8 software (GraphPad). Paired *t*-tests were performed to compare mIPSC frequency, mIPSC amplitude, mIPSC decay time, and evoked decay time before and after PTX addition. A non-linear regression was fitted to the block of eIPSCs and fIPSPs in PTX graphed by either time or stimulation number, and a sum-of-squares F test was used to determine if the curves were significantly different. One-way ANOVAs were used to compare initial eIPSC and fIPSP peak amplitude, time constants (τ) for different concentrations of external Ca$^{2+}$, and percent not blocked evoked signal. Two-way ANOVAs were used to compare paired pulse ratio in different concentrations of external Ca$^{2+}$. Tukey's post-hoc testing was used where appropriate following ANOVAs. Unpaired *t*-tests were used to compare mIPSC frequency and amplitude with the high Cl$^-$ pipette solution and paired *t*-tests were used to compare before and after TTX application. A Kolmogrov-Smirnov test was used to compare frequency distributions for mIPSC amplitude with standard and high Cl$^-$ pipette solution. A linear regression was used to calculate the rate of eIPSC recovery following block with PTX in high KCl. A sum-of-squares F test was used to determine if the slopes of these lines were significantly different. A non-linear regression was used to calculate the rate of recovery of mIPSCs following block with PTX in high KCl and muscimol, and a sum-of-squares F test was used to determine if the curves were significantly different.

Statistical significance was defined as p<0.05. Specifics of statistical tests employed are listed in figure legends.

## Acknowledgements

The authors would like to thank Dr. Ryan Hibbs for discussions at initial stages of this project. We would also like to thank Drs. Natali Chanaday, Ji-Woon Kim, Kanzo Suzuki, Melissa Bawa and PeiYi Lin and Robert Altamirano for comments during the preparation of this manuscript. This work was supported by National Institute of Health grants GM008203 (PMH), MH064913 (MKP), MH081060 and MH070727 (LMM), and MH66198 (ETK).

## Additional information

### Competing interests

Lisa M Monteggia: Reviewing editor, *eLife*. The other authors declare that no competing interests exist.

### Funding

| Funder | Grant reference number | Author |
| --- | --- | --- |
| National Institute of General Medical Sciences | T32 GM008203 | Patricia M Horvath |
| National Institute of Mental Health | R01 MH070727 | Lisa M Monteggia |
| National Institute of Mental Health | R01 MH66198 | Ege T Kavalali |
| National Institute of Mental Health | T32 MH064913 | Michelle K Piazza |

The funders had no role in study design, data collection and interpretation, or the decision to submit the work for publication.

### Author contributions

Patricia M Horvath, Conceptualization, Data curation, Formal analysis, Validation, Investigation, Visualization, Methodology, Project administration; Michelle K Piazza, Conceptualization, Data curation, Formal analysis, Methodology, Writing - review and editing; Lisa M Monteggia, Ege T Kavalali, Conceptualization, Supervision, Funding acquisition, Project administration

### Author ORCIDs

Patricia M Horvath  https://orcid.org/0000-0001-9969-8637
Michelle K Piazza  https://orcid.org/0000-0002-7852-3456
Lisa M Monteggia  https://orcid.org/0000-0003-0018-501X
Ege T Kavalali  https://orcid.org/0000-0003-1777-227X

### Ethics

Animal experimentation: Animal procedures conformed to the Guide for the Care and Use of Laboratory Animals and were approved by the Institutional Animal Care and Use Committee at UT Southwestern Medical Center ( Animal Protocol Number APN 2016-101416) and at Vanderbilt University School of Medicine (Animal Protocol Number M1800103).

### Decision letter and Author response

Decision letter https://doi.org/10.7554/eLife.52852.sa1
Author response https://doi.org/10.7554/eLife.52852.sa2

## Additional files

### Supplementary files
• Transparent reporting form

### Data availability
All source data files are included in the manuscript and supporting files.

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
