## [Decision Letter]

**Acceptance summary:**

The manuscript by Horvath et al. carefully documents a novel segregation of inhibitory synaptic receptors between those responsible for spontaneous (miniature) responses and those driving activity dependent release. This work builds on the power of dispersed ex vivo cultures of neurons that facilitates detailed physiological studies, and the exploitation of the fact that the channel blocking inhibitor picrotoxin can be used to provide information on recent use of receptors. Overall, these results show that similar to what is already known about excitatory synapses, inhibitory synapses show a similar subdivision into spontaneous vs evoked, although the segregation is less complete for inhibition.

**Decision letter after peer review:**

Thank you for sending your article entitled "Spontaneous and evoked neurotransmission are partially segregated at inhibitory synapses" for peer review at *eLife*. Your article is being evaluated by two peer reviewers, one of whom is a member of our Board of Reviewing Editors, and the evaluation is being overseen by Richard Aldrich as the Senior Editor.

There is potential interest in this paper, however there are two major concerns. The full reviews are posted below. By the way, the identification of picrotoxin as a use-dependent blocker is a nice finding.

Can you briefly respond to the following:

1) The potential impact of this report might be limited given that the extent of segregation between spontaneous and evoked GABA-containing vesicles seems to be much less than for excitatory vesicles. This combined with some results that were perceived to be ambiguous in their interpretation potentially further weaken the paper.

2) The results obtained are limited to one preparation (immature monolayer culture) and it is unclear how general this finding is in relation to developmental stage and in the "more intact" nervous system.

Reviewer #1:

Horvath et al. , using the model system of immature rat hippocampal dispersed cultures, report on the potential independence of GABA-a receptors mediating spontaneous vs activity-dependent synaptic release of the inhibitory neurotransmittter GABA. They find that the GABA-a receptor pore blocking antagonist picrotoxin shows novel use-dependence that should enable the use of this compound to infer information about release probabilities at inhibitory synapses, given that high release probabilities will lead to relatively greater release of GABA and therefore greater use-dependent block.

Using this approach, the authors show that, as previously shown for excitatory synaptic receptors, inhibitory receptors show partial segregation i.e. that the receptors mediating spontaneous (activity-independent, presumably spontaneous single vesicle release events) do no overlap 100% with those activated by evoked synaptic transmission.

The results are novel, in that segregation of receptors has not, to my knowledge, been reported for inhibitory synapses, and could be important in terms of further understanding of the mechanisms regulating synaptic transmission in the brain. The experiments are carefully executed and illustrated, and are for the most part convincing. A minor limitation is that the results are limited to this elegant, yet artificial cell culture system that may not fully recapitulate the native perisynaptic environment in the brain. This last issue could reduce the potential impact of the report.

1) Validation in another neural system, such as acute or cultured brain slices would broaden the interpretation and impact.

Reviewer #2:

In this manuscript Horvath al., described experiments performed on cultured rat hippocampal neurons performed to investigate if there is postsynaptic segregation of evoked and spontaneous release at inhibitory synapses. The work is an extension of earlier studies from the labs of Drs Kavalali and Monteggia which have studied how evoked and spontaneous release differ at many synapses in the hippocampus. I do not believe that the manuscript is suitable for publication in its current form because of a number of uncertainties concerning experimental design and interpretation of the findings. My major concern relates to interpretation the experiments described in Figure 4 after which the authors propose that some GABA receptors are only activated by spontaneous release and unaffected by concurrent evoked release. The design of this experiment was set up to preferentially increase the block of receptors involved in evoked release. The 90 sec exposure to picrotoxin and had previously been shown (Figure 3B) to only block 50% of the minis. However, following the application of 90 mM potassium and picrotoxin there appeared to be almost complete block of both spontaneous and evoked IPSCs. After application, the recovery of spontaneous transmission appeared more impaired than recovery of evoked transmission. I believe this result indicates that the authors have shown that some GABA receptors are involved in only evoked release and that the GABA receptors involved in spontaneous release are also involved in it evoked release. In addition to addressing this question I have other specific questions for the authors:

1) Please describe, in the Materials and methods or an early part of the Results, how eIPSCs were elicited? Was it by field stimulation or some other method? The time of decay of eIPSC changed with PTX and time. When was the exemplar in 1C recorded (how long after application) and was that steady state?

2) Please provide data on eIPSC decrement with stimulation/frequency in the absence of PTX (see Figure 1E). This will facilitate our interpretation of data comparing synaptic activity at different time points in later experiments.

3) Is the Y axis in Figure 3A (and 4E) measurement of normalized mIPSC frequency or a product of frequency and amplitude? If the latter, please justify.

4) There are a number of occasions in which it would be interesting to know how mIPSCs changed over time since "metabolic rundown" that affected eIPSCs in Figure 3 could also affect mIPSCs. In fact, in Figure 3B there appears to be ~30% rundown in 2 mins of basal activity in this set of experiments. This is surprising since Figure 3—figure supplement 1 shows frequency and amplitude are unaffected by TTX in unmatched experiments. The case would be stronger if it was demonstrated within recordings that TTX did not affect frequency or amplitude. Perhaps the experiments used to make the brown circle data in Figure 3F can be analyzed to test this idea.

[Editors' note: further revisions were suggested prior to acceptance, as described below.]

Thank you for submitting your revised article "Spontaneous and evoked neurotransmission are partially segregated at inhibitory synapses" for consideration by *eLife*. Your article has been reviewed by two peer reviewers, one of whom is a member of our Board of Reviewing Editors, and the evaluation has been overseen by Richard Aldrich as the Senior Editor. The reviewers have opted to remain anonymous.

The reviewers have discussed the reviews with one another and the Reviewing Editor has drafted this decision to help you prepare a revised submission.

Summary:

The manuscript by Horvath et al. nicely documents a novel segregation of inhibitory synaptic receptors between those responsible for spontaneous (miniature) responses and those driving activity dependent release. This work builds on the power of dispersed ex vivo cultures of neurons that facilitates detailed physiological studies, and the exploitation of the fact that the channel blocking inhibitor picrotoxin can be used to provide information on recent use of receptors. Overall, these results show that similar to what is already known about *excitatory* synapses, inhibitory synapses show a similar subdivision into spontaneous vs evoked, although the segregation is less complete for inhibition.

Overall, this is an important contribution to the literature, as it provides novel information of inhibitory signaling in the brain, which is critical in maintaining proper circuit balance. The study should be the starting point for further studies of mechanisms and consequences of such segregation.

The authors made the following significant changes to the manuscript:

1) They added field recordings from acute hippocampal slices and replicated key findings obtained from cell cultures.

2) They added new experiments using muscimol.

3) Investigated use-dependence of PTX unblock at 2 frequencies.

Revisions:

– The only remaining substantive critique is that the LFP approach for assessing inhibition is not a standard one, and more direct experiments utilizing intracellular experiments would certain be more compelling, although these are not strictly required.

– In addition, there are few, if any details regarding the precise methods of ensuring that the evoked LFP response accurately portrays inhibitory signaling. Missing details are the intensity of the stimulus, both in relative (% of maximal) and absolute (µA) values, number of trials averaged, confirmation of pharmacological dissection, rejection/inclusion criteria, etc.

– With the insertion of the new slice data in the manuscript, it is not clear whether the subsequent results were obtained in slices or dispersed culture.

– Whenever possible, include raw traces in every figure as examples.

– On Figure 5, the baseline of the evoked events should be visible.

– The muscimol experiments quite important, and they should be integrated into the main figures.

---

## [Author Response]

There is potential interest in this paper, however there are two major concerns. The full reviews are posted below. By the way, the identification of picrotoxin as a use-dependent blocker is a nice finding.Can you briefly respond to the following:1) The potential impact of this report might be limited given that the extent of segregation between spontaneous and evoked GABA-containing vesicles seems to be much less than for excitatory vesicles. This combined with some results that were perceived to be ambiguous in their interpretation potentially further weaken the paper.

Our results demonstrate that segregation of postsynaptic GABA receptors responding to evoked and spontaneous GABA release is not as complete as our previous reports on NMDA receptors and AMPA receptors (Atasoy et al., 2008; Sara et al., 2011). We think elucidating this fundamental property of GABAergic neurotransmission, regardless of the experimental outcome is extremely important. In our opinion, it was surprising that we could detect any segregation at all as there is limited evidence for biochemical signaling mediated by GABAergic transmission. Our earlier studies have associated segregation of evoked and spontaneous excitatory transmission with their distinct effects on biochemical signaling. Our current findings suggest a partial dissociation of putative signaling functions for evoked and spontaneous GABAergic neurotransmission, which we plan to pursue in future work.

Moreover, we believe the approach and the methods we demonstrate here will enable other groups to test the same question in their systems. For instance, our earlier work was instrumental in studies that were performed in the *Drosophila* neuromuscular junction by other groups (e.g. Peled et al., 2014).

2) The results obtained are limited to one preparation (immature monolayer culture) and it is unclear how general this finding is in relation to developmental stage and in the "more intact" nervous system.

We thank the reviewer for this suggestion. Our extensive work using hippocampal culture preparations show that synapses in this system reach maturity within two weeks as judged by their functional as well as ultrastructural properties (Kavalali et al., 1999; Mozhayeva et al., 2002). A key benefit of the culture systems is the ability to stimulate a large majority of the synaptic contacts onto the neuron of interest. Nevertheless, we have tested key predictions of these results in acute hippocampal slices and found a similar portion of the inhibitory evoked response remains after incubation with picrotoxin at rest to block GABA_A_Rs activated by spontaneous release. These results indicate that across systems 34-40% of the inhibitory evoked response is mediated by GABA_A_Rs which are only activated by evoked release and strengthen a key finding of our study.

Reviewer #1:Horvath et al. , using the model system of immature rat hippocampal dispersed cultures, report on the potential independence of GABA-a receptors mediating spontaneous vs activity-dependent synaptic release of the inhibitory neurotransmittter GABA. They find that the GABA-a receptor pore blocking antagonist picrotoxin shows novel use-dependence that should enable the use of this compound to infer information about release probabilities at inhibitory synapses, given that high release probabilities will lead to relatively greater release of GABA and therefore greater use-dependent block.Using this approach, the authors show that, as previously shown for excitatory synaptic receptors, inhibitory receptors show partial segregation i.e. that the receptors mediating spontaneous (activity-independent, presumably spontaneous single vesicle release events) do no overlap 100% with those activated by evoked synaptic transmission.The results are novel, in that segregation of receptors has not, to my knowledge, been reported for inhibitory synapses, and could be important in terms of further understanding of the mechanisms regulating synaptic transmission in the brain. The experiments are carefully executed and illustrated, and are for the most part convincing. A minor limitation is that the results are limited to this elegant, yet artificial cell culture system that may not fully recapitulate the native perisynaptic environment in the brain. This last issue could reduce the potential impact of the report.1) Validation in another neural system, such as acute or cultured brain slices would broaden the interpretation and impact.

We thank the reviewer for their kind words regarding our work and thoughtful suggestions. We have conducted additional experiments to test key predictions in hippocampal slice. Using this system and measuring fIPSPs we were able to recapitulate our findings from culture. As in culture, PTX acted in a use-dependent manner and was able to distinguish between different presynaptic release probabilities based on the rate of fIPSP block. Additionally, following incubation with PTX at rest to block GABA_A_Rs activated by spontaneous signaling, we found that 34 ± 5.37% of evoked response remains. This is remarkably similar to our results in culture, in which 40.1 ± 9.6% of the evoked response remained following incubation with PTX at rest.

Reviewer #2:In this manuscript Horvath al., described experiments performed on cultured rat hippocampal neurons performed to investigate if there is postsynaptic segregation of evoked and spontaneous release at inhibitory synapses. The work is an extension of earlier studies from the labs of Drs Kavalali and Monteggia which have studied how evoked and spontaneous release differ at many synapses in the hippocampus. I do not believe that the manuscript is suitable for publication in its current form because of a number of uncertainties concerning experimental design and interpretation of the findings. My major concern relates to interpretation the experiments described in Figure 4 after which the authors propose that some GABA receptors are only activated by spontaneous release and unaffected by concurrent evoked release. The design of this experiment was set up to preferentially increase the block of receptors involved in evoked release. The 90 sec exposure to picrotoxin and had previously been shown (Figure 3B) to only block 50% of the minis. However, following the application of 90 mM potassium and picrotoxin there appeared to be almost complete block of both spontaneous and evoked IPSCs. After application, the recovery of spontaneous transmission appeared more impaired than recovery of evoked transmission. I believe this result indicates that the authors have shown that some GABA receptors are involved in only evoked release and that the GABA receptors involved in spontaneous release are also involved in it evoked release.

We thank the reviewer for their insightful comments. In our experiments, 90 seconds of 55 mM of potassium was used to elicit a flood of GABA neurotransmitter that would activate GABA_A_Rs beyond those normally activated by evoked release. We believe the increased block of mIPSCs supports this premise. However, the reviewer makes a good point that 90 seconds of 55 mM of potassium may not be the best way to block all GABA_A_Rs, and may preferentially block those receptors which mediate responses to evoked release. Therefore, we have performed additional experiments utilizing the GABA_A_R agonist muscimol to activate and block GABA_A_Rs. Our experiments with muscimol recapitulate our findings with KCl, and indicate that this manipulation did not selectively block receptors activated by evoked release.

Our interpretation of the data in this figure relies on the recovery of the GABAergic response following PTX block being use-dependent (as shown by new data in the supplement and described above). If the receptors activated by evoked and spontaneous release are overlapping, we would expect stimulation to activate the same receptors as spontaneous release, thereby speeding up the recovery of the spontaneous mIPSC response. However, if the receptors activated by evoked and spontaneous release are non-overlapping, we would expect stimulation to activate a separate set of receptors and therefore have limited impact on the recovery of the spontaneous mIPSC response, which would instead rely only on spontaneous signaling.

Initially, using our non-standard measure of mIPSC recovery we did not find an impact of stimulation on the rate of mIPSC recovery and thus concluded this represented a set of receptors which was only activated by spontaneous signaling. Upon reanalysis of our data to use the more standard integrated charge as a measure of GABAergic recovery, we did find that stimulation increased the rate of mIPSC recovery. This is more in line with our previous block experiments showing partial overlap of the GABA_A_Rs activated by spontaneous and evoked release. However, this data does not exclude the possibility that the recovery of mIPSCs in the absence of stimulation is partly mediated by receptors which are only activated by spontaneous release. The slow recovery of mIPSCs in the absence of stimulation is likely due to the slow rate of spontaneous release at any given synapse (~1 vesicle/minute) compared to the 0.2 Hz applied stimulation (1 stimulation/5 seconds).

In addition to addressing this question I have other specific questions for the authors:1) Please describe, in the Materials and methods or an early part of the Results, how eIPSCs were elicited? Was it by field stimulation or some other method? The time of decay of eIPSC changed with PTX and time. When was the exemplar in 1C recorded (how long after application) and was that steady state?

We thank the reviewer for their careful examination of our study. The eIPSCs were elicited by bath stimulation through parallel platinum electrodes. The exemplar in 1C was taken from the 10^th^ response to stimulation in PTX following 8 minutes of PTX application at rest (no stimulation). We have clarified this point in the text.

2) Please provide data on eIPSC decrement with stimulation/frequency in the absence of PTX (see Figure 1E). This will facilitate our interpretation of data comparing synaptic activity at different time points in later experiments.

We thank the reviewer for this insightful point. We agree that this will be useful to plot these data together and have adjusted our figures accordingly. As seen in Figure 1E, the rundown associated with stimulation in the absence of PTX is far less than the block of GABAergic response measured in the presence of PTX.

3) Is the Y axis in Figure 3A (and 4E) measurement of normalized mIPSC frequency or a product of frequency and amplitude? If the latter, please justify.

The Y-axes in these figures is a normalized product of mIPSC frequency and amplitude. We initially chose this metric as a measure of charge transfer because the integrated charge of each event would be affected by decreased decay time following PTX addition. This would decrease overall charge transferred without necessarily providing a full measure of receptor block. However, we recognize that this is a non-standard measure and have updated our analysis to reflect integrated charge.

4) There are a number of occasions in which it would be interesting to know how mIPSCs changed over time since "metabolic rundown" that affected eIPSCs in Figure 3 could also affect mIPSCs. In fact, in Figure 3B there appears to be ~30% rundown in 2 mins of basal activity in this set of experiments. This is surprising since Figure 3—figure supplement 1 shows frequency and amplitude are unaffected by TTX in unmatched experiments. The case would be stronger if it was demonstrated within recordings that TTX did not affect frequency or amplitude. Perhaps the experiments used to make the brown circle data in Figure 3F can be analyzed to test this idea.

The reviewer makes an excellent point and we have included experiments in which spontaneous IPSCs were measured within recordings before and after TTX. These recordings show that the frequency and amplitude of events is unchanged following the addition of TTX indicating that we are measuring bona fide mIPSCs. Additionally, mIPSCs are stable in TTX over the course of 10 minutes indicating that they are not affected by the same rundown which effects evoked IPSCs.

[Editors' note: further revisions were suggested prior to acceptance, as described below.]

Revisions:– The only remaining substantive critique is that the LFP approach for assessing inhibition is not a standard one, and more direct experiments utilizing intracellular experiments would certain be more compelling, although these are not strictly required.

While this may not be a standard approach, it allowed us to test the prediction of our findings in mature hippocampal slices. We have followed as closely as possible guidelines from other published works measuring inhibition in hippocampal slice (Younts et al., 2016). The ability of picrotoxin to block this response further indicates that it is a measureable inhibitory response. Unfortunately, due to institutional restrictions, we are not able to perform additional experiments at this time. However, we appreciate the critique and in future work, we will definitely test the premise of this study using whole cell voltage clamp methods in distinct GABAergic synapses in the CNS.

– In addition, there are few, if any details regarding the precise methods of ensuring that the evoked LFP response accurately portrays inhibitory signaling. Missing details are the intensity of the stimulus, both in relative (% of maximal) and absolute (µA) values, number of trials averaged, confirmation of pharmacological dissection, rejection/inclusion criteria, etc.

We have added these details and others to our Materials and methods section on hippocampal slice electrophysiology.

– With the insertion of the new slice data in the manuscript, it is not clear whether the subsequent results were obtained in slices or dispersed culture.

We have updated the text to provide clarity as to whether experiments in each section were performed in culture or slices.

– Whenever possible, include raw traces in every figure as examples.

We have now included example traces in all main figures and most supplemental figures.

– On Figure 5, the baseline of the evoked events should be visible.

We have adjusted our example traces in this figure to include the baseline both before and after the evoked events. For clarity, the stimulus artifact was removed from these traces, as well as those in Figure 6.

– The muscimol experiments quite important, and they should be integrated into the main figures.

We have moved this data into a newly created figure, Figure 6, to better highlight these results.